# EV-Associated miRNAs from Peritoneal Lavage are a Source of Biomarkers in Endometrial Cancer

**DOI:** 10.3390/cancers11060839

**Published:** 2019-06-18

**Authors:** Berta Roman-Canal, Cristian Pablo Moiola, Sònia Gatius, Sarah Bonnin, Maria Ruiz-Miró, Esperanza González, Xavier González-Tallada, Ivanna Llordella, Isabel Hernández, José M. Porcel, Antonio Gil-Moreno, Juan M. Falcón-Pérez, Julia Ponomarenko, Xavier Matias-Guiu, Eva Colas

**Affiliations:** 1Department of Pathology and Molecular Genetics/Oncologic Pathology Group, Biomedical Research Institute of Lleida (IRBLleida), University of Lleida, CIBERONC, 25198 Lleida, Spain; bertaromancanal@gmail.com (B.R.-C.); cristian.pablo@vhir.org (C.P.M.); sgatius.lleida.ics@gencat.cat (S.G.); mruiz@irblleida.cat (M.R.-M.); fjgonzalezt.lleida.ics@gencat.cat (X.G.-T.); ivanna.llordella@gmail.com (I.L.); Isabelmedico@gmail.com (I.H.); 2Department of Pathology, University Hospital of Bellvitge, Bellvitge Biomedical Research Institute (IDIBELL), 08908 L’Hospitalet de Llobregat, Spain; 3Biomedical Research Group in Gynecology, Vall Hebron Research Institute (VHIR), CIBERONC, 08035 Barcelona, Spain; antonioimma@yahoo.es; 4Universitat Autònoma de Barcelona (UAB), 08193 Bellaterra, Spain; 5Centre for Genomic Regulation (CRG), The Barcelona Institute or Science and Technology, Dr. Aiguader 88, 08003 Barcelona, Spain; sarah.bonnin@crg.eu (S.B.); julia.ponomarenko@crg.eu (J.P.); 6Exosomes Laboratory and Metabolomics Platform. CIC bioGUNE, CIBEREHD Bizkaia Technology Park, 48160 Derio, Spain; egonzalez@cicbiogune.es (E.G.); jfalcon@cicbiogune.es (J.M.F.-P.); 7Pleural Medicine Unit, Arnau de Vilanova University Hospital, Cancer Biomarker Group, IRBLleida, 25198 Lleida, Spain; jporcelp@yahoo.es; 8Gynecological Oncology Department, Vall Hebron University Hospital, CIBERONC, 08035 Barcelona, Spain; 9IKERBASQUE, Basque Foundation for Science, 48011 Bilbao, Spain; 10University Pompeu Fabra, 08002 Barcelona, Spain

**Keywords:** endometrial cancer, uterine cancer, exosomes, biomarkers, miRNAs, ascitic fluid, peritoneal lavage, liquid biopsy, extracellular vesicles, microRNAs

## Abstract

Endometrial cancer (EC) is the sixth most common cancer in women worldwide and is responsible for more than 89,000 deaths every year. Mortality is associated with presence of poor prognostic factors at diagnosis, i.e., diagnosis at an advanced stage, with a high grade and/or an aggressive histology. Development of novel approaches that would permit us to improve the clinical management of EC patients is an unmet need. In this study, we investigate a novel approach to identify highly sensitive and specific biomarkers of EC using extracellular vesicles (EVs) isolated from the peritoneal lavage of EC patients. EVs of peritoneal lavages of 25 EC patients were isolated and their miRNA content was compared with miRNAs of EVs isolated from the ascitic fluid of 25 control patients. Expression of the EV-associated miRNAs was measured using the Taqman OpenArray technology that allowed us to detect 371 miRNAs. The analysis showed that 114 miRNAs were significantly dysregulated in EC patients, among which eight miRNAs, miRNA-383-5p, miRNA-10b-5p, miRNA-34c-3p, miRNA-449b-5p, miRNA-34c-5p, miRNA-200b-3p, miRNA-2110, and miRNA-34b-3p, demonstrated a classification performance at area under the receiver operating characteristic curve (AUC) values above 0.9. This finding opens an avenue for the use of EV-associated miRNAs of peritoneal lavages as an untapped source of biomarkers for EC.

## 1. Introduction

Endometrial cancer (EC) is the sixth most common cancer in women worldwide and is responsible for more than 89,000 deaths every year [1]. EC is predominantly a disease that afflicts postmenopausal women, occurs in women older than 50 years in more than 90% of cases, and is detected at a mean age of 65 [2]. Approximately 10% of cases are diagnosed in premenopausal women, 5% of whom are younger than 40 years. Mortality is associated with presence of poor prognostic factors at diagnosis, i.e., diagnosis at an advanced stage, with a high grade and/or an aggressive histology. Patients presenting any of those features are at increased risk of recurrence, and, for them, therapeutic options are limited. Although most ECs are diagnosed early, up to 10% of tumors are diagnosed at a late stage, where the five-year survival drops to 16% compared to 95% in women diagnosed at an early stage [3]. Regarding histology, 80% of EC patients are diagnosed with an endometrioid histology with the average 5-year survival of 75%. Nonetheless, the 20% of patients diagnosed with a non-endometrioid histology account for 47% of EC-related deaths. Grade 3 endometrioid tumors are diagnosed in 15% of all EC patients, although theses tumors are responsible for 27% of EC-related deaths.

The cornerstone treatment of EC is surgery, which is mostly standardized through all hospitals with slight variations. The national comprehensive cancer network (NCCN) recommends both surgical and pathological staging with total hysterectomy, bilateral salpingo-oophorectomy, and peritoneal cytology. Although lymphadenectomy still remains the most reliable way of avowing downstaging and correctly identifying patients who require adjuvant chemotherapy or radiotherapy, recent data have questioned its role in early stage EC due to the high variability in node involvement and the frequently associated comorbidities. Lymphadenectomy might not be performed in early stage EC patients in some centers. Moreover, fertility-sparing surgery (FSS) in reproductive-age patients affected by endometrial cancer has attracted attention in the last decade because the consequences of an approach that is too radical may have a severe impact on a patient’s quality of life and psychological well-being [4].

The new era of molecular advances has fostered biomarker research, although the identification of useful biomarkers in liquid biopsies remains a challenge. Among all serum biomarkers, the human epididymis protein 4 (HE4) has been one of the most investigated in EC. HE4 was found to be sufficiently specific but poorly sensitive in patients with EC. The diagnostic performance of HE4 appears to be better than that of the cancer antigen 125 (CA125) in diagnosing EC at an early stage, but its real value and efficacy for management of EC have not been clearly demonstrated in clinical practice [5]. Hence, a clinical challenge in EC is the development of novel molecular approaches to liquid biopsies that permit early diagnosis and recurrence control.

MicroRNAs (miRNAs) are small, non-coding RNA molecules of about 22 nucleotides in length [6] that regulate gene expression at the post-transcriptional level by inhibiting protein translation or destabilizing target transcripts via binding to the 3’-untranslated region (3’UTR), resulting in transcriptional repression or mRNA degradation upon dicer cleavage [7]. miRNAs have been found to play a critical role in almost every physiological process, including differentiation, proliferation, and apoptosis. They have also been described as oncogenes or tumor suppressors in some tumors [8], including EC [6]. Although miRNAs are detected intracellularly, they pass into the extracellular space and can be detected in a broad variety of bodily fluids, either freely in circulation or contained in extracellular vesicles (EVs) [9].

There are different types of EVs, and their size ranges from 20 to 200 nm. The largest vesicles, i.e., microvesicles, are released directly from the budding of the plasma membrane. The smallest vesicles, i.e., exosomes, are formed within intracellular multivesicular bodies and released by their fusion with the cellular membrane. Their function is to mediate intercellular communication, influencing the recipient cell’s behavior. Importantly, EVs have attracted the interest of the scientific community as a source of biomarkers, mainly because they carry a broad range of bioactive material (proteins, metabolites, RNA, miRNA, etc.) that is well-protected by the lipid bilayer membrane of EVs, even if they are extracted from circulating or proximal bodily fluids or frozen before any experimental study [10].

Herein, we investigate the use of EVs isolated from the peritoneal lavage, a proximal fluid of EC, as a source of potential EC biomarkers. The peritoneal lavage, at surgery, just before starting the manipulation of the uterus, was used for staging purposes according to the old International Federation of Gynecology and Obstretrics (FIGO) staging rules. In several centers, peritoneal washing is still performed because of the prognostic information that the presence of cancer cells provides by cytologic examination. However, this fluid has not been used for molecular analysis. In this study, we conducted miRNA profiling of EVs isolated from the peritoneal lavages of 25 EC patients and the ascitic fluids of 25 non-cancer patients using the TaqMan OpenArray Human MicroRNA Panel. We identified the most relevant individual miRNAs related to EC and characterized the biological and molecular landscape of the EC milieu. The study was conceived as a proof of concept investigation to demonstrate the feasibility of using the peritoneal lavage as a source of EV-associated miRNA biomarkers of EC.

## 2. Results

We analyzed the miRNA profile of EVs isolated from the ascitic fluids of 25 control patients and the peritoneal lavages of 25 EC patients. Figure 1 illustrates the workflow that was followed in this study.

The quality of EVs isolated from the ascitic fluids and peritoneal lavages was measured by size distribution and concentration by Nanoparticle Tracking Analysis. The analysis demonstrated that we analyzed a population that was mostly comprised of small EVs but that also contained a low number of microvesicles. The EVs isolated from EC and non-EC patients did not differ in concentration and size (Appendix A). miRNAs were extracted from all EVs and they were used for a systematic miRNA expression analysis using the Taqman OpenArray technology. We detected 371 out of the 754 miRNAs (49.2%) present in the OpenArray. Probes that had a Cycle threshold (Ct) value of 40 in all samples and samples in which more than 80% of the probes had a Ct value greater than 40 were removed, resulting in a study that contained a total of 355 miRNAs from 22 control and 22 EC patients (Table 1).

The differential expression analysis between cancer and control cases yielded a list of 114 miRNAs that were significantly dysregulated (adj. *p* < 0.05 and abs(logFC) > = 1). Among those, 96 miRNA were found to be downregulated and 18 miRNA were upregulated in EC patients (Table 2). To evaluate whether these miRNAs can be used as biomarkers, we performed a predictive analysis using the logistic modeling. Eight miRNAs demonstrated predictive performance with area under the receiver operating characteristic curve (AUC) values above 0.90, including miRNA-383-5p, miRNA-10b-5p, miRNA-34c-3p, miRNA-449b-5p, miRNA-34c-5p, miRNA-200b-3p, miRNA-2110, and miRNA-34b-3p (Table 2, rows in bold; Figure 2). All eight miRNAs were significantly downregulated in EC patients (from 3.75-fold to 12.18-fold in the log2_scale).

To further understand the milieu generated by EVs in the context of EC, we performed a bioinformatics study to first identify the predicted transcripts that are regulated by all of the differentially expressed miRNAs and then to assess the biological processes and molecular functions that they participate in. A total of 8074 transcripts were found to be regulated by the 114 differentially expressed miRNAs. To comprehensively integrate the properties of all target transcripts, they were classified with the Gene Ontology (GO) terms shown in Figure 3A,B.

## 3. Discussion

In this study, we investigated for the first time the miRNA content of EVs isolated from peritoneal lavages and ascitic fluids of EC and control patients, respectively. Our study shows that the EV-associated miRNAs can be consistently extracted from those proximal bodily fluids and that the miRNA expression profiles can indicate and represent the status of EC patients. The EV-associated miRNAs were analyzed using the Taqman OpenArray technology. The differential expression analysis yielded 114 miRNAs that were significantly dysregulated in EC patients.

An abundance of scientific research has been published regarding the role of miRNAs in EC [11]. Torres et al. published the first study focused on miRNA expression both in tissue and plasma samples of EC patients. They investigated the expression of miRNA-99a, miRNA-100, and miRNA-199b, which target the mTOR kinase. A combined signature of miRNA-99a and miRNA-199b in plasma samples resulted in 88% sensitivity and 93% specificity, indicating a good diagnostic potential [12]. Despite these findings, they were not applied in the clinical setting [13]. In this respect, EVs arise as a source of biomarkers with an unexploited potential. They can be isolated from bodily fluids, such as saliva, blood, urine, malignant pleural effusion, and ascitis [14]. In EC, miRNAs isolated from EVs have been scarcely studied. Akhil et al. evaluated the potential of the miRNA content of urine-derived EVs as a diagnostic biomarker in EC patients [15], and Hanzi Xu et al. isolated EVs from serum samples and identified 209 upregulated and 66 downregulated circular RNA (circRNAs) in EVs from serum of patients with EC compared with those from serum of healthy controls [16].

Although plasma, serum, and urine biopsies are the most common liquid biopsies, the use of proximal bodily fluids as a source of biomarkers has attracted the attention of the biomarker research community. Proximal bodily fluids, such as uterine fluid for EC, offer an improved representation of the molecular alterations that take place in the tumor [17]. The peritoneal fluid is another proximal fluid of EC; however, this type of proximal fluid has not been yet exploited to investigate EC-related biomarkers or any other cancer originating within the peritoneal cavity.

To the best of our knowledge, our study is the first to report the value of this proximal fluid for the identification of miRNAs associated with EVs in EC. Importantly, this study identified the dysregulation of 114 miRNAs, among which miRNA-383-5p, miRNA-10b-5p, miRNA-34c-3p, miRNA-449b-5p, miRNA-34c-5p, miRNA-200b-3p, miRNA-2110, and miRNA-34b-3p are of special interest, as they demonstrated a high classification potential. Interestingly, some of these miRNAs were found in previous EC studies. In concordance with our study, miRNA-10b and miRNA-34b were found to be downregulated in endometrial serous adenocarcinoma versus normal endometrial tissue, indicating that these miRNAs might also be associated with the aggressive subtype of the serous EC. In fact, in that study, reduced miRNA-10b expression was found to be significantly correlated with shorter overall survival [18]. MiRNA-34 has been described as a fundamental regulator of tumor suppression; it controls multiple protein targets involved in the cell cycle and apoptosis, and was associated with metastasis and chemoresistance [19]. In contrast to our study, miRNA-200b was found to be upregulated in endometrial serous adenocarcinoma versus normal endometrial tissue [18]. However, it has been reported to be downregulated in various human malignancies, and its function has been postulated to be oncogenic (i.e., involved in proliferation, motility, apoptosis, stemness, and the epithelial-to-mesenchymal transition) [20]. Interestingly, accumulating evidence in the field of endometriosis suggests that apoptosis that occurs in the peritoneal cavity may play a pivotal role in addressing the immune homeostasis in the peritoneal microenvironment [21]. This causes scavenging mechanisms to fail, allowing for the survival of endometriotic cells in patients with endometriosis [22]. In the context of endometrial cancer, we speculate that EVs derived from endometrial cancer cells might reach the peritoneal cavity and target the mesothelial liner cells of the peritoneum and ovaries to modulate the immune homeostasis and create a more favorable milieu to metastasize. In fact, most of the metastasis associated with endometrial cancer occurs either in the vagina, in the lymph nodes, or within the peritoneal cavity. Several of the miRNAs identified in our study are related to tumor progression in endometrial cancer or in other tumor types. It is possible that endometrial cancer cells may spread into the peritoneum and the ovaries through the Fallopian tube, and develop peritoneal metastasis and ovarian metastasis in the absence of lympho-vascular space invasion. It is important to note that normal endometrium may spread into the peritoneum and the ovaries through retrograde menstruation, and give rise to endometriosis. Metastasis that originates through this pathway may be associated with indolent behavior. A subset of ECs from ovarian metastasis are associated with such a good prognosis that they were interpreted as synchronous tumors [23]. Next-generation sequence analysis has confirmed that they are metastatic. Assessing transtubal exosomal release from endometrial cancer may help us to understand the mechanisms involved in this type of indolent metastasis.

The study leaves some questions open. This study enabled us to identify a large number of dysregulated miRNAs associated with EVs in the peritoneal lavage of EC patients. We think that the EVs that we have identified come from endometrial and mesothelial cells in the EC group, and mostly from mesothelial cells in the control group. The comparative analysis of EC patients with non-cancer patients with ascites suggests that these selected miRNAs come from EVs of EC tissue. However, there is obviously the probability that a subset of EVs came from an inflammatory reaction associated with EC and some other factors distinct from the EC pathology, such as the source of EVs (peritoneal lavage vs. ascitic fluids), the surgery (after induction of general anesthesia), and the control samples, which were obtained by means of paracentesis (with local anesthesia). Nevertheless, these promising biomarkers should be further validated as well as combined in order to increase the already excellent accuracy of each individual miRNA. This should be done in an independent study involving a larger cohort of EC patients versus a control group with a higher biologic variability, including, for example, patients with leiomyomas or women requiring tubal ligation for definitive contraception. Although we tested whether or not differentially expressed miRNAs were dependent on gender, further studies should include only female controls. Moreover, further research should be directed to an evaluation of the prognostic potential of each specific dysregulated miRNA, as this might help to guide the surgical treatment of EC patients.

## 4. Materials and Methods

### 4.1. Patients and Ascitic Fluid and Peritoneal Lavage Collection

All subjects provided informed consent before they participated in the study. The study was conducted in accordance with the Declaration of Helsinki, and the protocol was approved by The Clinical Research Ethics Committee of Hospital Arnau de Vilanova in Lleida, Spain (Approval number: CEIC-1630). Samples were obtained with support from the IRBLleida Biobank (B.0000682) and Plataforma biobancos PT17/0015/0027. Ascitic fluids and peritoneal lavages were extracted from a cohort of 50 patients, corresponding to 25 control patients with decompensated cirrhosis and 25 patients with EC who underwent curative surgery. In the control patients, the collection of ascitic fluids was performed as follows: Ascitic fluids were aspirated using 18 or 21G needles (for diagnostic paracentesis) or an over-the-needle catheter device (for therapeutic paracentesis). The procedure was performed under sterile conditions, the needle insertion site was selected by ultrasound guidance, and the skin and parietal peritoneum were previously anesthetized with 2% mepivacine. A total of 100 mL of ascitic fluid was gently aspirated, collected into a 50 mL tube, and stored at −80 °C until use. In EC patients, the collection of peritoneal lavage was performed during surgery, once the abdominal cavity was opened and prior to any manipulation of the uterus. A total of 100 mL of physiological saline was instilled into the abdominal cavity with a 50 mL syringe, mobilizing patients for the correct distribution of saline, which was then extracted with a 50 mL syringe connected to a 14-gauge aspiration needle. The peritoneal lavage was gently aspirated. A volume ranging from 50 to 100 mL was collected and stored at −80 °C until use. The clinical features of each patient are listed in Appendix A.

### 4.2. EV Isolation

EVs were isolated with a differential centrifugation method as previously described [24] with slight modifications. Briefly, ascitic fluids and peritoneal lavages were centrifuged at 300× *g* for 10 min, followed by centrifugation at 2500× *g* for 20 min at the moment that the sample was collected, and frozen at −80 °C. Then, samples were centrifuged at 10,000 *g* for 30 min (Thermo Scientific Heraeus MultifugeX3R Centrifuge (FiberLite rotor F15-8x-50c)). The supernatant was filtered through 0.22 µm filters (Merck Millipore), and the obtained sample was transferred to ultracentrifuge tubes (Beckman Coulter), which were filled with phosphate-buffered saline (PBS), to perform two consecutive ultracentrifugation steps at 100,000 *g* for 2 hours each on a Thermo Scientific Sorvall WX UltraSeries Centrifuge with an AH-629 rotor. The pellet containing the EVs was resuspended in 50 µL of PBS. From those, 5 µL were isolated for nanoparticle tracking analysis (NTA) and quantification, and the rest was frozen at −80 °C with 500 µL of Qiazol for RNA extraction.

### 4.3. Nanoparticle Tracking Analysis

The size and number of EVs were determined using a Nanosight LM10 instrument equipped with a 405 nm laser and a Hamamatsu C11440 ORCA-Flash 2.8 camera (Hamamatsu) with Nanoparticle Tracking Analysis (NTA, Malvern Instruments, UK). Each sample was diluted appropriately with Milli-Q water (Milli-Q Synthesis, Merck Millipore, MA, USA) to give counts in the linear range of the instrument. The particles in the laser beam underwent Brownian motion, and a video was recorded for 60 s in triplicate. The analysis was performed by following the manufacturer’s instructions, and data were analyzed using version 2.3 of the NTA software.

### 4.4. Total RNA Extraction and OpenArray Analysis

Total RNA, including miRNAs and other RNAs, was isolated from the EV samples using the miRNeasy MiniKit (Qiagen, Hilden, Germany) according to the manufacturer’s protocol. RNA from EVs was eluted with 30 µL of Nuclease-free water (Qiagen, Hilden, Germany). MiRNA expression was determined using a TaqMan OpenArray Human MicroRNA Panel, QuantStudio 12K Flex (Catalog number: 4470187, Thermo Fisher Scientific, Waltham, MA, USA), a fixed-content panel containing 754 well-characterized human miRNA sequences from the Sanger miRBase v14, and according to the manufacturer’s instructions. Reverse transcription (RT) was performed on 2 µL RNA using Megaplex™ Primer Pools A and B and the supporting TaqMan^®^ MicroRNA Reverse Transcription Kit as follows: 15 min at 42 °C and 5 min at 85 °C. Then, 5 µL of the resulting cDNA was preamplified prior to real-time PCR analysis using Megaplex™ PreAmp Pools and the TaqMan^®^ PreAmp Master Mix using the following conditions: one single step at 95 °C for 5 min, 20 cycles of a two-step program (3 sec, 95 °C and 30 sec, 60 °C) followed by a single cycle of 10 min at 99 °C to inactivate the enzyme. The preamplified products were diluted to 1:20 in 0.1× TE buffer pH 8.0, and mixed in a 1:1 ratio with TaqMan^®^ OpenArray^®^ Real-Time PCR Master Mix in the 384-well OpenArray^®^ Sample Loading Plate. TaqMan^®^ OpenArray^®^ MicroRNA Panels were automatically loaded using the AccuFill™ System.

### 4.5. Preprocessing and Differential Expression Analysis

All bioinformatics analyses were performed with BioConductor (version 3.7) [25] in the R statistical environment (version 3.5.0) [26]. For the data preprocessing, the HTqPCR (version 1.34) R package [27] was used. Probes that had a “Cycle threshold” (Ct) value of 40 in all samples were removed. Samples in which more than 80% of the probes had a Ct value greater than 40 were retained. To assure comparability across samples, the Ct values were delta-normalized. The average Ct values of the probes hsa−miR−150−5p, hsa−let−7g-5p, hsa−miR−598−3p, and hsa−miR−361−3p were used for normalization. These probes had Ct values of 40 in a maximum of three samples and the lowest interquartile range across all samples. The differential expression analysis was carried out with the empirical Bayes approach on linear models using the limma (version 3.36) R Package [28]. Results were corrected for multiple testing using the False Discovery Rate (FDR) [29].

### 4.6. Development of Predictors

For predictive analysis, the whole patient cohort was randomly divided into training and validation sets with a ratio of 3:2. Calculated (with the limma R Package) relative miRNA expression values were used as input variables into a logistic regression model between groups. Each miRNA (adjusted *p*-value <0.05) was fitted in the logistic regression model to differentiate the EC and the control patient groups in the training set, and its classification ability was evaluated using the area under the ROC curve (AUC), accuracy, sensitivity, and specificity values on the validation set. The procedures for the division of the patient cohort into training and validation sets and fitting the logistic model were repeated 500 times and statistics were collected.

### 4.7. miRNA Target Gene Prediction and Bioinformatics Analysis

miRNA target genes were predicted using the Predictive Target Module of the miRWalk2.0 online software [30] (https://goo.gl/ajG9ja). Only genes for which miRNAs recognize a minimum 7 bp seed length, seed start at position 1, and sequence localize at 3´UTR were considered as valid targets. Moreover, to improve target gene prediction accuracy, we considered only those transcripts that were predicted in at least eight out of the 12 databases (miRWalk, miRanda, MicroT4, miRDB, miRMap, miRBridge, miRNAMap, PICTAR2, RNA22, PITA, TargetScan, and RNAhybrid) presented in the miRWalk2.0 tool.

The online Panther software [31] (http://www.pantherdb.org/) was used for the Gene Ontology (GO) functional analysis to analyze the potential functions of the predicted target genes. Biological process (BP) and molecular function (MF) GO terms were analyzed and plotted.

## 5. Conclusions

Thanks to this study, we have demonstrated that the use of EV-associated miRNAs of ascitic fluid from control patients and peritoneal lavages from EC patients are an untapped source of biomarkers. Specifically, we identified 114 dysregulated miRNAs, and, among those, miRNA-383-5p, miRNA-10b-5p, miRNA-34c-3p, miRNA-449b-5p, miRNA-34c-5p, miRNA-200b-3p, miRNA-2110, and miRNA-34b-3p were highlighted as promising biomarkers of EC with an AUC value higher than 0.90.

## Figures and Tables

**Figure 1 cancers-11-00839-f001:**
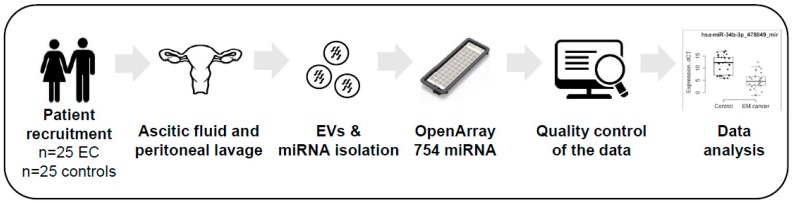
Workflow of the study design. Abbreviations: EC, Endometrial Cancer; EVs, Extracellular vesicles.

**Figure 2 cancers-11-00839-f002:**
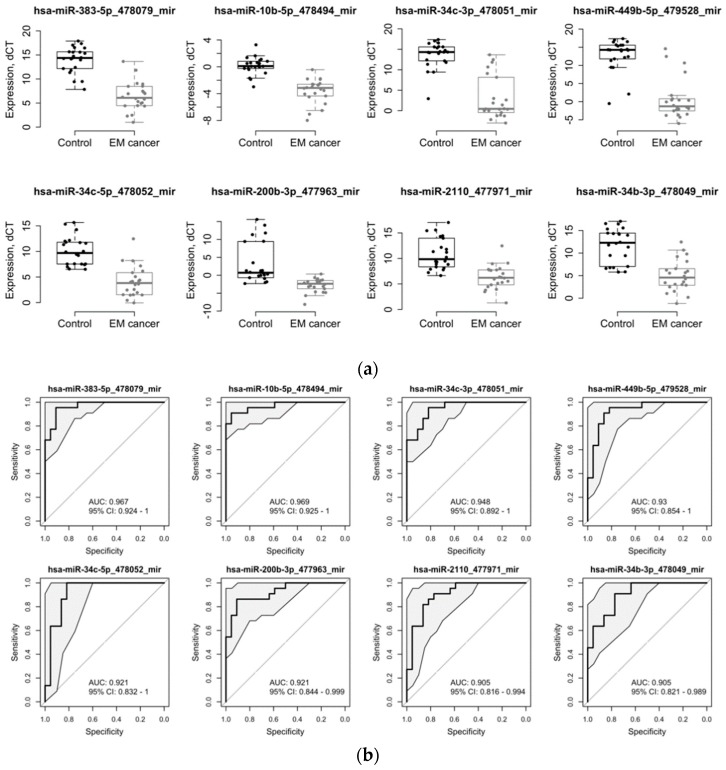
Diagnostic performance of the top eight differentially expressed miRNAs: (**a**) Relative dCT values of the top differentially expressed miRNAs in patients with EC (*n* = 22) compared to control patients (*n* = 22), ** *p* < 0.05; (**b**) receiver operating characteristic (ROC) curves and AUC scores for the top eight differentially expressed miRNAs.

**Figure 3 cancers-11-00839-f003:**
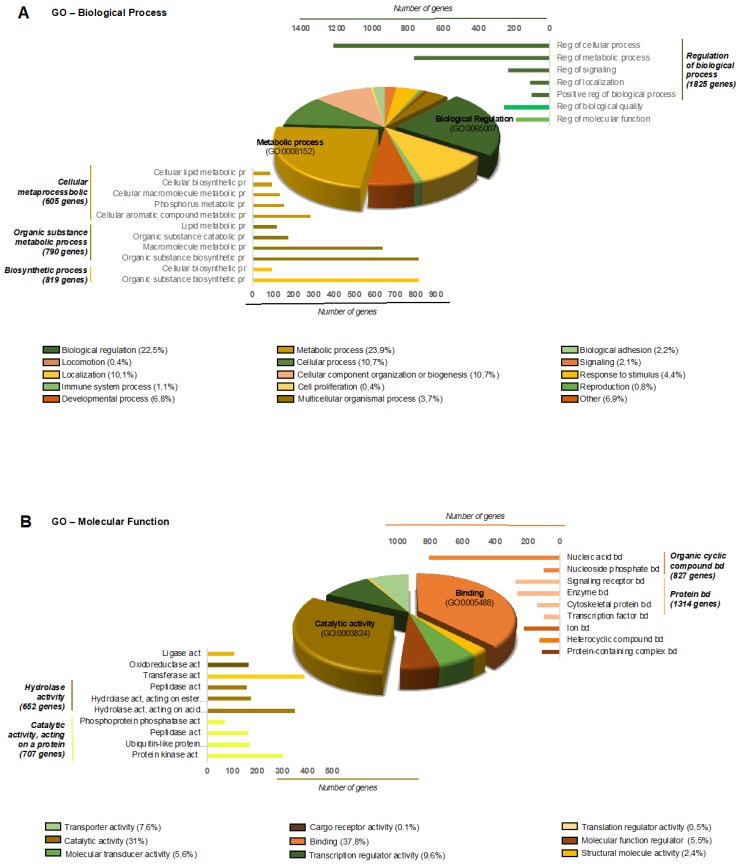
Gene Ontology (GO) analysis of the predicted proteins regulated by miRNA from EVs of EC and control patients: (**A**) Biological process GO analysis of predicted proteins regulated by differentially expressed miRNA from the ELV of EC and control patients. A total of 8074 genes were predicted to be modulated by 114 miRNA and were included in at least one of the GO Biological process categories as indicated in the pie chart. Others include the following categories that have less than 0.4% of representation: growth; multi-organism process; biological phase; rhythmic process; pigmentation; nitrogen utilization. (**B**) Molecular Function GO analysis of predicted proteins regulated by differentially expressed miRNA from the ELV of EC and control patients. A total of 8074 genes were predicted to be modulated by 115 miRNA and were included in at least one of the GO Molecular Function categories as indicated in the pie chart. Abbreviations: Reg, Regulation; Pr, Process; Bd, Binding; Act, Activity.

**Table 1 cancers-11-00839-t001:** Clinicopathological characteristics of patients.

Clinical Parameters	Endometrial Cancer	Control
Age		
Median	61.4	65.5
Minimum	46	52
Maximum	78	90
Gender		
Female	22	4
Male	-	18
Pathology		
Endometrial Cancer		-
EEC	19	-
NEEC	3	-
Hepatic cirrhoses	-	20
Other	-	22

Clinical characteristics of the final cohort of patients included in the study after data normalization.

**Table 2 cancers-11-00839-t002:** List of MicroRNA (miRNA) transcripts displaying a significant differential expression in patients with EC compared to control patients.

miRNA	logFC	*p*-Value	adj.*p*-Value	AUC	AUC_95%CI_Lower	AUC_95%CI_Upper	Accuracy	Sensitivity	Specificity
**hsa-miR-383-5p_478079_mir**	**−7.56**	**2.86E-14**	**1.02E-11**	**0.961**	**0.938**	**0.985**	**0.884**	**0.876**	**0.892**
**hsa-miR-10b-5p_478494_mir**	**−3.75**	**2.62E-11**	**3.10E-09**	**0.958**	**0.93**	**0.987**	**0.897**	**0.864**	**0.93**
**hsa-miR-34c-3p_478051_mir**	**−10.56**	**9.57E-10**	**6.79E-08**	**0.948**	**0.925**	**0.97**	**0.843**	**0.893**	**0.794**
**hsa-miR-449b-5p_479528_mir**	**−12.18**	**2.58E-11**	**3.10E-09**	**0.927**	**0.896**	**0.959**	**0.864**	**0.896**	**0.83**
**hsa-miR-34c-5p_478052_mir**	**−5.97**	**5.60E-10**	**4.97E-08**	**0.924**	**0.891**	**0.957**	**0.845**	**0.86**	**0.83**
**hsa-miR-200b-3p_477963_mir**	**−6.10**	**2.73E-06**	**6.91E-05**	**0.917**	**0.888**	**0.946**	**0.846**	**0.824**	**0.869**
**hsa-miR-2110_477971_mir**	**−4.57**	**7.27E-07**	**2.58E-05**	**0.906**	**0.875**	**0.938**	**0.802**	**0.748**	**0.855**
**hsa-miR-34b-3p_478049_mir**	**−6.73**	**4.17E-07**	**2.12E-05**	**0.903**	**0.874**	**0.932**	**0.752**	**0.719**	**0.786**
hsa-miR-200c-3p_478351_mir	−10.29	3.42E-09	2.02E-07	0.897	0.863	0.931	0.799	0.744	0.853
hsa-miR-150-5p_477918_mir	2.89	9.74E-06	2.26E-04	0.892	0.856	0.928	0.861	0.908	0.814
hsa-miR-1180-3p_477869_mir	−4.62	9.40E-07	3.04E-05	0.89	0.854	0.926	0.791	0.738	0.844
hsa-miR-29c-5p_478005_mir	−2.39	1.35E-06	3.70E-05	0.883	0.849	0.918	0.773	0.773	0.774
hsa-miR-190a-5p_478358_mir	−2.51	1.42E-03	9.18E-03	0.871	0.831	0.911	0.816	0.813	0.82
hsa-miR-708-5p_478197_mir	−5.51	1.19E-06	3.52E-05	0.865	0.826	0.903	0.77	0.684	0.856
hsa-miR-218-5p_477977_mir	−6.70	1.27E-05	2.59E-04	0.859	0.816	0.901	0.715	0.53	0.9
hsa-miR-99a-3p_479224_mir	−3.23	4.78E-05	7.78E-04	0.858	0.818	0.897	0.732	0.733	0.73
hsa-miR-196b-5p_478585_mir	−6.63	5.15E-07	2.29E-05	0.85	0.805	0.894	0.766	0.624	0.908
hsa-miR-142-5p_477911_mir	8.86	1.89E-05	3.35E-04	0.849	0.808	0.889	0.748	0.898	0.599
hsa-miR-193a-3p_478306_mir	−2.49	1.31E-03	8.93E-03	0.849	0.807	0.891	0.789	0.853	0.725
hsa-miR-193a-5p_477954_mir	−3.10	1.02E-05	2.26E-04	0.848	0.808	0.888	0.791	0.78	0.802
hsa-miR-181a-5p_477857_mir	−1.71	1.01E-04	1.49E-03	0.847	0.807	0.887	0.755	0.831	0.678
hsa-miR-125b-5p_477885_mir	−1.86	5.22E-05	8.06E-04	0.845	0.803	0.887	0.744	0.784	0.704
hsa-miR-429_477849_mir	−3.07	2.17E-04	2.65E-03	0.843	0.797	0.89	0.792	0.817	0.767
hsa-miR-30a-3p_478273_mir	−1.72	4.82E-05	7.78E-04	0.841	0.803	0.88	0.698	0.738	0.657
hsa-miR-196a-5p_478230_mir	−6.80	1.89E-05	3.35E-04	0.837	0.801	0.873	0.743	0.803	0.682
hsa-miR-34b-5p_478050_mir	−7.34	6.87E-07	2.58E-05	0.835	0.787	0.884	0.795	0.855	0.735
hsa-miR-769-5p_478203_mir	−2.81	1.60E-03	1.01E-02	0.833	0.788	0.878	0.768	0.724	0.812
hsa-miR-20a-5p_478586_mir	−1.74	2.60E-04	2.88E-03	0.832	0.789	0.875	0.727	0.738	0.716
hsa-miR-320a_478594_mir	−8.30	8.68E-04	6.84E-03	0.831	0.796	0.867	0.698	0.59	0.806
hsa-miR-30d-5p_478606_mir	−2.36	1.01E-02	3.59E-02	0.827	0.785	0.869	0.728	0.746	0.709
hsa-miR-409-3p_478084_mir	3.24	2.24E-04	2.65E-03	0.822	0.777	0.868	0.779	0.872	0.687
hsa-miR-30b-5p_478007_mir	−2.57	1.65E-03	1.01E-02	0.82	0.777	0.863	0.714	0.736	0.693
hsa-miR-598-3p_478172_mir	−1.84	3.68E-04	3.53E-03	0.82	0.778	0.861	0.723	0.677	0.77
hsa-miR-369-3p_478067_mir	4.94	1.32E-04	1.81E-03	0.816	0.772	0.86	0.709	0.895	0.523
hsa-miR-187-3p_477941_mir	−6.40	1.10E-04	1.56E-03	0.815	0.768	0.861	0.749	0.638	0.86
hsa-miR-210-3p_477970_mir	−1.85	1.37E-04	1.81E-03	0.813	0.768	0.859	0.699	0.713	0.686
hsa-miR-29c-3p_479229_mir	−2.84	1.63E-03	1.01E-02	0.812	0.773	0.851	0.742	0.749	0.735
hsa-miR-885-5p_478207_mir	−3.98	3.50E-04	3.45E-03	0.812	0.768	0.857	0.713	0.741	0.686
hsa-miR-142-3p_477910_mir	3.77	2.85E-04	2.98E-03	0.81	0.764	0.857	0.71	0.775	0.645
hsa-miR-342-3p_478043_mir	2.17	5.34E-03	2.44E-02	0.808	0.763	0.854	0.721	0.739	0.703
hsa-miR-17-3p_477932_mir	−6.78	2.76E-04	2.97E-03	0.806	0.762	0.85	0.739	0.826	0.652
hsa-miR-92a-3p_477827_mir	−1.49	5.35E-04	4.87E-03	0.805	0.763	0.848	0.711	0.796	0.626
hsa-miR-29a-5p_478002_mir	−6.21	2.46E-03	1.34E-02	0.803	0.762	0.843	0.681	0.802	0.561
hsa-miR-30e-3p_478388_mir	−4.45	6.43E-03	2.78E-02	0.796	0.749	0.844	0.633	0.426	0.839
hsa-miR-125a-5p_477884_mir	−1.39	1.17E-03	8.33E-03	0.794	0.749	0.839	0.746	0.768	0.723
hsa-miR-130a-3p_477851_mir	−1.76	4.73E-04	4.42E-03	0.793	0.744	0.842	0.723	0.836	0.61
hsa-miR-485-3p_478125_mir	2.34	6.14E-03	2.72E-02	0.793	0.746	0.841	0.694	0.692	0.698
hsa-miR-223-3p_477983_mir	1.80	6.62E-04	5.73E-03	0.79	0.744	0.837	0.714	0.756	0.672
hsa-miR-21-5p_477975_mir	1.63	7.79E-04	6.55E-03	0.787	0.738	0.836	0.714	0.706	0.721
hsa-miR-194-5p_477956_mir	−1.29	7.80E-03	3.18E-02	0.785	0.735	0.835	0.749	0.746	0.752
hsa-miR-29a-3p_478587_mir	−2.40	8.07E-03	3.22E-02	0.784	0.736	0.833	0.663	0.746	0.581
hsa-miR-887-3p_479189_mir	−5.02	2.54E-04	2.88E-03	0.784	0.736	0.832	0.652	0.46	0.843
hsa-miR-135a-5p_478581_mir	−5.63	7.93E-04	6.55E-03	0.775	0.727	0.823	0.698	0.663	0.732
hsa-miR-31-5p_478015_mir	−2.16	8.14E-04	6.56E-03	0.775	0.727	0.823	0.671	0.769	0.573
hsa-miR-29b-2-5p_478003_mir	−2.82	8.96E-04	6.92E-03	0.773	0.723	0.824	0.671	0.688	0.654
hsa-miR-26b-5p_478418_mir	−1.47	1.01E-03	7.45E-03	0.771	0.722	0.82	0.669	0.822	0.516
hsa-miR-656-3p_479137_mir	4.06	1.73E-03	1.04E-02	0.771	0.72	0.822	0.693	0.7	0.686
hsa-miR-33b-5p_478479_mir	−4.09	1.48E-02	4.82E-02	0.767	0.718	0.816	0.64	0.591	0.69
hsa-miR-141-3p_478501_mir	−8.97	1.31E-05	2.59E-04	0.765	0.715	0.815	0.718	0.847	0.589
hsa-miR-423-3p_478327_mir	−1.80	3.02E-04	3.07E-03	0.765	0.714	0.816	0.679	0.67	0.687
hsa-miR-28-3p_477999_mir	−2.33	2.24E-03	1.28E-02	0.764	0.714	0.813	0.658	0.621	0.695
hsa-miR-29b-3p_478369_mir	−2.44	6.09E-04	5.40E-03	0.761	0.715	0.808	0.684	0.63	0.737
hsa-let-7b-5p_478576_mir	−1.43	1.22E-03	8.51E-03	0.76	0.715	0.806	0.624	0.648	0.6
hsa-miR-551b-3p_478159_mir	−4.13	2.14E-03	1.25E-02	0.76	0.711	0.809	0.671	0.751	0.591
hsa-miR-154-3p_478725_mir	3.28	2.78E-03	1.47E-02	0.759	0.704	0.814	0.673	0.738	0.608
hsa-miR-18a-3p_477944_mir	−2.84	7.96E-03	3.21E-02	0.759	0.707	0.81	0.67	0.696	0.644
hsa-miR-26a-5p_477995_mir	−1.33	6.71E-03	2.87E-02	0.759	0.709	0.809	0.685	0.622	0.748
hsa-miR-181d-5p_479517_mir	−5.43	4.75E-03	2.31E-02	0.757	0.709	0.805	0.654	0.524	0.784
hsa-miR-151a-3p_477919_mir	−1.65	1.05E-03	7.59E-03	0.756	0.701	0.811	0.731	0.794	0.667
hsa-miR-449a_478561_mir	−5.77	2.18E-04	2.65E-03	0.755	0.701	0.808	0.712	0.812	0.611
hsa-miR-219a-5p_477980_mir	−3.58	2.31E-03	1.30E-02	0.752	0.704	0.8	0.686	0.708	0.664
hsa-miR-584-5p_478167_mir	−5.62	2.55E-03	1.37E-02	0.752	0.703	0.801	0.662	0.778	0.545
hsa-miR-20b-5p_477804_mir	−4.91	3.96E-03	2.04E-02	0.747	0.698	0.796	0.674	0.784	0.563
hsa-miR-24-3p_477992_mir	−1.30	5.35E-03	2.44E-02	0.747	0.696	0.798	0.665	0.688	0.642
hsa-miR-33a-5p_478347_mir	−3.72	2.83E-03	1.48E-02	0.742	0.689	0.795	0.672	0.586	0.757
hsa-let-7e-5p_478579_mir	−6.66	4.07E-03	2.06E-02	0.741	0.685	0.797	0.709	0.794	0.623
hsa-miR-543_478155_mir	2.69	1.40E-03	9.18E-03	0.741	0.687	0.795	0.685	0.544	0.826
hsa-miR-126-5p_477888_mir	5.31	1.01E-03	7.45E-03	0.74	0.686	0.795	0.723	0.826	0.62
hsa-miR-139-5p_478312_mir	−3.63	5.34E-03	2.44E-02	0.738	0.69	0.787	0.641	0.433	0.849
hsa-miR-652-3p_478189_mir	−1.13	1.11E-02	3.75E-02	0.737	0.687	0.786	0.613	0.621	0.605
hsa-miR-200a-3p_478490_mir	−4.89	6.25E-03	2.74E-02	0.735	0.682	0.788	0.709	0.79	0.629
hsa-miR-21-3p_477973_mir	2.34	8.87E-03	3.42E-02	0.735	0.679	0.791	0.689	0.738	0.641
hsa-miR-181c-5p_477934_mir	−1.86	1.08E-02	3.73E-02	0.734	0.68	0.789	0.678	0.659	0.698
hsa-miR-361-5p_478056_mir	−1.20	4.86E-03	2.31E-02	0.727	0.677	0.777	0.688	0.75	0.626
hsa-miR-296-5p_477836_mir	−4.15	4.87E-03	2.31E-02	0.726	0.673	0.778	0.631	0.536	0.726
hsa-miR-1271-5p_478674_mir	−4.54	1.01E-02	3.59E-02	0.725	0.674	0.776	0.643	0.783	0.502
hsa-miR-125b-2-3p_478666_mir	−3.58	9.83E-03	3.58E-02	0.724	0.672	0.776	0.637	0.761	0.514
hsa-miR-130b-3p_477840_mir	−4.14	4.77E-03	2.31E-02	0.723	0.673	0.774	0.684	0.66	0.707
hsa-miR-545-3p_479002_mir	−3.15	1.06E-02	3.69E-02	0.723	0.67	0.775	0.648	0.526	0.77
hsa-miR-331-3p_478323_mir	−5.14	7.33E-03	3.10E-02	0.722	0.669	0.775	0.673	0.753	0.594
hsa-miR-128-3p_477892_mir	−1.40	2.39E-03	1.33E-02	0.72	0.667	0.772	0.642	0.647	0.638
hsa-miR-148b-3p_477824_mir	−1.40	7.60E-03	3.17E-02	0.72	0.662	0.778	0.663	0.746	0.58
hsa-miR-30a-5p_479448_mir	−5.80	1.43E-02	4.70E-02	0.716	0.658	0.774	0.654	0.716	0.592
hsa-miR-222-3p_477982_mir	−1.46	1.25E-02	4.20E-02	0.712	0.655	0.77	0.653	0.645	0.661
hsa-miR-451a_478107_mir	−4.53	1.40E-03	9.18E-03	0.711	0.654	0.768	0.61	0.613	0.606
hsa-miR-203a-3p_478316_mir	−4.92	9.36E-03	3.50E-02	0.708	0.653	0.764	0.664	0.804	0.524
hsa-miR-758-3p_479166_mir	3.56	9.37E-03	3.50E-02	0.705	0.653	0.756	0.652	0.692	0.612
hsa-miR-452-5p_478109_mir	−2.16	1.11E-02	3.75E-02	0.703	0.644	0.762	0.598	0.571	0.626
hsa-miR-505-3p_478145_mir	−1.91	6.03E-03	2.71E-02	0.703	0.647	0.759	0.604	0.553	0.655
hsa-miR-504-5p_478144_mir	−2.25	1.06E-02	3.69E-02	0.701	0.647	0.756	0.624	0.58	0.667
hsa-miR-182-5p_477935_mir	−5.24	9.84E-03	3.58E-02	0.7	0.639	0.761	0.686	0.683	0.688
hsa-miR-551a_478158_mir	−4.09	1.58E-02	4.93E-02	0.7	0.644	0.756	0.639	0.772	0.507
hsa-miR-101-3p_477863_mir	−2.79	4.16E-03	2.08E-02	0.699	0.645	0.752	0.626	0.417	0.835
hsa-miR-31-3p_478012_mir	−3.59	8.29E-03	3.27E-02	0.699	0.647	0.751	0.675	0.718	0.632
hsa-miR-140-3p_477908_mir	−2.87	1.56E-02	4.90E-02	0.697	0.642	0.753	0.622	0.439	0.805
hsa-miR-214-5p_478768_mir	−3.30	8.46E-03	3.30E-02	0.696	0.641	0.751	0.649	0.707	0.591
hsa-miR-199b-5p_478486_mir	−4.83	1.49E-02	4.82E-02	0.695	0.646	0.744	0.623	0.631	0.614
hsa-miR-337-5p_478036_mir	3.17	1.27E-02	4.22E-02	0.688	0.631	0.745	0.711	0.9	0.523
hsa-miR-154-5p_477925_mir	2.07	9.88E-03	3.58E-02	0.687	0.631	0.742	0.657	0.773	0.541
hsa-miR-582-5p_478166_mir	−3.50	1.52E-02	4.83E-02	0.681	0.618	0.743	0.644	0.702	0.587
hsa-miR-126-3p_477887_mir	5.32	1.82E-03	1.08E-02	0.678	0.616	0.739	0.738	0.832	0.643
hsa-miR-143-3p_477912_mir	−1.54	9.06E-03	3.46E-02	0.674	0.616	0.731	0.677	0.745	0.608
hsa-miR-324-5p_478024_mir	−1.34	1.52E-02	4.83E-02	0.651	0.588	0.714	0.572	0.603	0.542
hsa-miR-145-5p_477916_mir	−1.70	7.80E-03	3.18E-02	0.649	0.591	0.708	0.646	0.723	0.57

Log fold-change expression, *p*-value, adjusted *p*-value, area under the receiver operating characteristic curve (AUC) values, accuracy, sensitivity, specificity, and 95% confidence intervals of the 114 dysregulated miRNAs.

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
