# Peer review of "EV-Associated miRNAs from Peritoneal Lavage are a Source of Biomarkers in Endometrial Cancer"

_cancers, 2019, doi:10.3390/cancers11060839_

Round 1

Reviewer 1 Report

The authors have done a good study to find eight significant miRNAs in the peritoneal lavage of EC patients, which show significant sensitivity in ROC-curve analysis.

The major control samples (~80%) in the study are from the male individual which authors have taken into consideration for comparison of control vs patient cases. This might mislead results.

How these differentially expressed miRNAs are validated? Whether the expression was tested in peritoneal lavage of patients not involved in the study or in hysterectomy and tumor samples of EC patients.

The authors should consider another supportive experiment such as RNA-ISH to show the expression of these miRNAs in FFPE uterine tissue sections from EC patients to validate the results.

Author Response

Attached please see our responses to Reviewer 1's comments.

Reviewer 2 Report

This paper brings novel information about the miRNA in the context of extracellular vesicles obtained by centrifugation after a peritoneal lavage to qualify endometrial cancer. 

This approach is however a bit complex and invasive and the justification to proceed to this lavage is already linked to the possibility of performing surgery. Therefore it seems very complicated to apply the method to diagnose endometrial cancer, which as recognized by the authors is identified at late stage in 10% of the patients. 

Seeing the bright side of the method, the authors did obtain quite clear information with satisfactory specificities and sensibilities analyzed with their best 8 miRNAs. 

The authors end their study by analyzing the putative regulation of mRNA trnascripts mediated by the deregulated miRNA, this being based exclusively on bioinformatics. Since the discussion about the gene ontology is not made, and since the validity of the identification of the pathways need to be confirmed by direct biological approaches for some of the targets, I think that they should not be detailed as much in the paper. In addition, the sense of variation of the miRNA could also be taken into account in order to better understand the possible effects on miRNA expression. Also since these miRNA are in extracellular vesicles, it should be explained where (in which cells) are their putative targets, and whether the changes may have a potential effects on these cells. 

Author Response

Point 1: This paper brings novel information about the miRNA in the context of extracellular vesicles obtained by centrifugation after a peritoneal lavage to qualify endometrial cancer. This approach is however a bit complex and invasive and the justification to proceed to this lavage is already linked to the possibility of performing surgery. Therefore it seems very complicated to apply the method to diagnose endometrial cancer, which as recognized by the authors is identified at late stage in 10% of the patients. 

Response 1: Particularly in endometrial cancer (EC), the use of proximal bodily fluids, such as the uterine aspirates, has been very fruitful to identify diagnostic biomarkers with high sensitivity and specificity [1], [2]. Nevertheless, another important gap is the identification of prognostic, predictive and treatment-response biomarkers to control EC recurrence, which is directly associated to EC mortality. This problem was highlighted in the manuscript (line 86-87). In this work, we wanted to unveil the potential of the peritoneal fluid, another proximal bodily fluid, with a particular focus on understanding if this fluid is able to provide with sensitive and specific biomarkers of EC. We agree with the reviewer comment regarding the importance of identifying biomarkers in the least invasive manner for patients and we understand that the peritoneal fluid is not easily obtained before surgery. However, this fluid can be easily access during surgery and if it represents the molecular alterations occurring in EC, this fluid might be useful to identify prognostic, predictive or treatment response biomarkers. Our work permitted us to unveil that the peritoneal fluid is an excellent source of biomarkers and it represents the alterations found in EC. Among those, we believed that some of them might also be related to EC prognosis (information included in the discussion section). We strongly believe that this study is relevant to the scientific community since the use of this fluid in EC biomarker research was not previously reported, and it might open new avenues, with limitations for diagnosis, but it might be important for prognostic purposes. 

References

1.         Martinez-Garcia, E.; Lesur, A.; Devis, L.; Cabrera, S.; Matias-Guiu, X.; Hirschfeld, M.; Asberger, J.; van Oostrum, J.; Casares de Cal, M. de los Á.; Gómez-Tato, A.; et al. Targeted Proteomics Identifies Proteomic Signatures in Liquid Biopsies of the Endometrium to Diagnose Endometrial Cancer and Assist in the Prediction of the Optimal Surgical Treatment. Clinical Cancer Research 2017, 23, 6458–6467.

2.         Martinez-Garcia, E.; Lesur, A.; Devis, L.; Campos, A.; Cabrera, S.; van Oostrum, J.; Matias-Guiu, X.; Gil-Moreno, A.; Reventos, J.; Colas, E.; et al. Development of a sequential workflow based on LC-PRM for the verification of endometrial cancer protein biomarkers in uterine aspirate samples. Oncotarget 2016, 7.

Point 2: Seeing the bright side of the method, the authors did obtain quite clear information with satisfactory specificities and sensibilities analyzed with their best 8 miRNAs. The authors end their study by analyzing the putative regulation of mRNA transcripts mediated by the deregulated miRNA, this being based exclusively on bioinformatics. Since the discussion about the gene ontology is not made, and since the validity of the identification of the pathways need to be confirmed by direct biological approaches for some of the targets, I think that they should not be detailed as much in the paper. In addition, the sense of variation of the miRNA could also be taken into account in order to better understand the possible effects on miRNA expression.

Response 2: We agree with the reviewer that this part of the results could be removed. We have deleted this section from the manuscript.

Point 3: Also since these miRNA are in extracellular vesicles, it should be explained where (in which cells) are their putative targets, and whether the changes may have a potential effects on these cells.

Response 3: The peritoneal milieu is composed by mainly mesothelial linner cells in the peritoneum and the ovaries. Thus, in a context of EC, we believe that extracellular vesicles (EVs) derived from EC cells might reach the peritoneal cavity and target all those cells to create a more favorable milieu to metastasize. In fact, most of the metastasis associated to EC occur either in the vagina, lymph nodes or within the peritoneal cavity; and several of the miRNAs identified in our study are related to tumor progression in EC or in other type of tumors. We have included this explanation in line 226 (discussion section).

Reviewer 3 Report

I read with great interest the Manuscript titled “EV-associated miRNAs from peritoneal lavage is a source of biomarkers in endometrial cancer” (cancers-505924), which falls within the aim of Cancers.      

In my honest opinion, the topic is interesting enough to attract the readers’ attention. Methodology is accurate and conclusions are supported by the data analysis. Nevertheless, authors should clarify some points and improve the discussion citing relevant and novel key articles about the topic.

Authors should consider the following recommendations:

Manuscript should be further revised by a native English speaker.

The authors have not adequately highlighted the strengths and limitations of their study. I suggest better stressing these points.

 What are the actual clinical implications of this study? it is important to report the results obtained by the authors in the context of clinical practice and to adequately highlight what contribution this study adds to the literature already existing on the topic and to future study perspectives.

Authors have appropriately underlined that miRNAs plays a critical role in almost every physiological process such as differentiation, proliferation and apoptosis. In this regard, I suggest discussing, at least briefly, how apoptosis induction may play a key role in other diseases, referring to: PMID: 27628753; PMID: 28571791.

Authors identified 114 dysregulated miRNAs, and among those, miRNA-383-5p, miRNA310 10b-5p, miRNA-34c-3p, miRNA-449b-5p, miRNA-34c-5p, miRNA-200b-3p, miRNA-2110 and miRNA-34b-3p were highlighted as promising biomarkers of endometrial cancer. In this regard, I would add few lines to discuss the clinical management of early stage endometrial cancer, especially in reproductive-aged women: PMID: 30379327; PMID: 28108938; PMID: 28188573.

Author Response

Point 1: I read with great interest the Manuscript titled “EV-associated miRNAs from peritoneal lavage is a source of biomarkers in endometrial cancer” (cancers-505924), which falls within the aim of Cancers. In my honest opinion, the topic is interesting enough to attract the readers’ attention. Methodology is accurate and conclusions are supported by the data analysis. Nevertheless, authors should clarify some points and improve the discussion citing relevant and novel key articles about the topic.

Authors should consider the following recommendations:

Manuscript should be further revised by a native English speaker.

The authors have not adequately highlighted the strengths and limitations of their study. I suggest better stressing these points.

What are the actual clinical implications of this study? it is important to report the results obtained by the authors in the context of clinical practice and to adequately highlight what contribution this study adds to the literature already existing on the topic and to future study perspectives.

Response 1: The results that have been obtained in this work may have important implications for the appropriate managing of endometrial cancer (EC) patients, once they have been properly validated. We summarize here the main expected impacts & limitations:

1.- It is true that the molecular analysis of peritoneal washings cannot replace methods that are more easily performed, such as transvaginal ultrasound, magnetic resonance and endometrial biopsy. Thus, our main focus should not be to develop a diagnostic tool but to impact on the identification of highly sensitivity and specific biomarkers of EC that contained prognostic, predictive or treatment-response information. The current study was designed as a proof of concept to demonstrate that peritoneal washings may be a source of EC associated biomarkers. We are now increasing the series of cases and including their clinical follow up in order to be able to stratify the patients according to the miRNA-associated extracellular vesicles (EVs) profile in correlation with prognosis. However, this was not the aim of this first study. In here, we needed to first prove that the EVs contained in the peritoneal fluid are in fact a good source of tumor-derived miRNAs biomarkers.

2.- There is a second interest in the study. There are many evidences showing that tumor-derived EVs play a role in creating the appropriate microenvironment for distant sites, susceptible to receive tumor metastasis. Endometrial carcinoma usually spread through blood and lymphatic vessels, and metastasis are associated with adverse prognosis. There is, however, great interest in the possibility that EC cells may spread into the peritoneum and the ovaries through the Fallopian tube, and develop peritoneal metastasis and ovarian metastasis in the absence of lympho-vascular space invasion. Metastasis originated through this pathway may be associated with indolent behavior. A subset of ovarian metastasis from endometrial carcinomas are associated with such a good prognosis that they were interpreted a synchronous tumors. Next generation sequence analysis confirms that they are metastatic. Assessing transtubal exosomal release from EC may help in understanding the mechanisms involved in such type of indolent metastasis. We can expand comments on these aspects in the discussion if this is requested by the reviewers.

Point 2: Authors have appropriately underlined that miRNAs plays a critical role in almost every physiological process such as differentiation, proliferation and apoptosis. In this regard, I suggest discussing, at least briefly, how apoptosis induction may play a key role in other diseases, referring to: PMID: 27628753; PMID: 28571791.

Response 2: We agree with the reviewers that these publications will add value to our discussion. Thus, the following paragraph has been modified including the underlined text: “MiRNA-34 has been described as a fundamental regulator of tumor suppression, controlling multiple protein targets involved in the cell cycle and apoptosis and was associated to metastasis and chemoresistance [17]. Controversial with our study, miRNA-200b was found to be up-regulated in endometrial serous adenocarcinoma vs normal endometrial tissue [16] although its downregulation was reported in various human malignancies and its function has been postulated as oncogenic (i.e. involved in proliferation, motility, apoptosis, stemness, and epithelial-to-mesenchymal transition) [18]. Interestingly, accumulating evidence in the field of endometriosis has already suggested that apoptosis occurring in the peritoneal cavity may play a pivotal role in addressing the immune homeostasis in the peritoneal microenvironment [21]. This causes the failure of scavenging mechanisms, allowing the survival of endometriotic cells in patients with endometriosis, but might also influence tumor progression within the peritoneal cavity and the onset of tumor metastasis in a context of EC [22].”

Point 3: Authors identified 114 dysregulated miRNAs, and among those, miRNA-383-5p, miRNA310 10b-5p, miRNA-34c-3p, miRNA-449b-5p, miRNA-34c-5p, miRNA-200b-3p, miRNA-2110 and miRNA-34b-3p were highlighted as promising biomarkers of endometrial cancer. In this regard, I would add few lines to discuss the clinical management of early stage endometrial cancer, especially in reproductive-aged women: PMID: 30379327; PMID: 28108938; PMID: 28188573.

Response 3: We agree with the reviewer that we should emphasize more the clinical implications of our results in the management of EC patients. Thus, we have included the following modifications in introduction (line 70 to 73 and line 83 to 93) and discussion (line 240).

Reviewer 4 Report

The manuscript “EV-associated miRNAs from peritoneal lavage is a source of biomarkers in endometrial cancer” (cancers-505924) evaluates the potential of peritoneal lavage during surgery in patients suffering from endometrial cancer as a source of biomarkers, in terms of EV-associated miRNAs. Authors found 8 miRNAs (namely miRNA-383-5p, miRNA-10b-5p, miRNA-34c-3p, miRNA-449b-5p, miRNA-34c-5p, miRNA-200b-3p, miRNA-2110 and miRNA-34b-3p), which demonstrated a classification performance at AUC values above 0.9 for patients with endometrial cancer. The paper is well written and English is appropriate.

Nevertheless, I have several concerns regarding the study design employed. Although the authors state in the abstract (lines 51-52) and introduction (lines 70-71) that mortality is associated with poor prognostic factors, the analyses conducted are intended to employ EV-associated miRNAs as a source of diagnostic biomarkers. Particularly in endometrial cancer, nowadays image techniques (transvaginal ultrasound and magnetic resonance) permit a high accurate diagnostic performance. Under suspicion of endometrial malignancy, a biopsy of endometrial tissue is obtained to perform the histological examination, which together with image techniques results (myometrial infiltration, locoregional extension…) determines the indication for either surgical or medical treatment. Therefore, the use of EV-associated miRNAs obtained by peritoneal lavage during surgery for curative treatment of endometrial cancer might not be the best source for diagnostic biomarkers in this condition. Additionally, the control cohort the authors employ is inappropriate for several reasons: 1) they compare peritoneal lavage in patients to ascitic fluids in control subjects (note that the hypoalbuminemia consecuence of the hepatic cirrhosis might dramatically alter the content of released material into the peritoneal cavity, in contrast with the content observed in peritoneal lavages); 2) patients samples are obtained during surgery (after induction of general anaesthesia) and control samples are obtained by means of paracentesis (with local anaesthesia); 3) patient group is composed by women and control group is mainly composed by men. Furthermore, no data regarding body mass index, associated medical treatment, histological endometrial cancer subtypes, etc. is provided.

Regarding the methodological approach, a validation (by RT-qPCR) of the results found in the exploratory phase (TaqMan OpenArray Human MicroRNA Panel) is expected, which unfortunately lacks in this study. Regarding EVs isolation, authors describe storing peritoneal lavages and ascites at -80ºC and, latterly, isolating EVs. On so doing, the centrifugation steps previous to the ultracentrifugation lose their sense, since bigger vesicles might have been destroyed by freezing.

Regarding results section, several analyses that might improve the quality of the paper might need to be considered. Not overlooking the limited sample size, further analyses (preferably with RT-qPCR values) regarding presence/absence of metastasis and endometrioid endometrial cancer and non-endometrioid endometrial cancer and survival/exitus or time to recurrence are needed. In this sense, a molecular fingerprint in terms of miRNA content could improve the management of the patients, an improvement needed in the field, as stated by the authors in the introduction section.

The study, as it is, seems to provide 8 miRNAs that distinguish endometrial cancer patients from hepatic cirrhosis patients, which is not clinically very sound.

Additionally, in the discussion section I miss a rationale for searching extracellular vesicles in endometrial cancer, considering that only three out of the 25 patients have peritoneal metastasis. Might these EVs be produced as part of inflammation? If this is the case, could they be considered as valuables biomarkers of endometrial cancer or perhaps they are also found in peritoneal lavages in other pathologies?

Additionally, there are some minor mistakes that might need to be solved, in the eventual publication of the work:

Line 91: the concept of “double-stranded mRNA” might not be completely right in this context.

Figure 1: displays the image of “colon cancer” versus control.  

Line 273: “pH8” is not appropriately written.

Line 299: The authors might need to review the definition “miRNA seed start at position 1” according to the current literature.

Therefore, after carefully reviewing the manuscript, I cannot recommend the paper for publication in the “Cancers” journal in its current state.

Author Response

Attached please see our responses to Reviewer 4's comments.

Round 2

Reviewer 2 Report

The authors responded perfectly well to my expectations.

Author Response

Thank you very much for your valuable comments.

Reviewer 4 Report

Dear Editors of the “Cancers” journal:

The manuscript “EV-associated miRNAs from peritoneal lavage is a source of biomarkers in endometrial cancer” (cancers-505924) evaluates the potential of peritoneal lavage during surgery in patients suffering from endometrial cancer as a source of biomarkers, in terms of EV-associated miRNAs. Authors found 8 miRNAs (namely miRNA-383-5p, miRNA-10b-5p, miRNA-34c-3p, miRNA-449b-5p, miRNA-34c-5p, miRNA-200b-3p, miRNA-2110 and miRNA-34b-3p), which demonstrated a classification performance at AUC values above 0.9 for patients with endometrial cancer. The paper is well written and English is appropriate.

After my initial comments, authors have provided more detailed information, explanations and performed the majority of suggested corrections regarding minor mistakes, which altogether have improved the quality of the paper.

On the one hand, a major limitation of the study in its initial form was the way in which samples had been processed, this is, freezing before centrifugations. Now that authors have clarified the right methodology, I have no further concerns in this point, which renders results more reliable.

In their responses, the authors provided evidence that miRNA expression into their results was not affected by gender. Additionally, authors have provided further information for the patient cohort regarding histological grade and subtype, medical treatment, primary tumor localization and metastasis, which improved the quality of the paper.

The authors highlight that the main objective of the paper is to demonstrate that peritoneal lavage is a source of EVs-related miRNAs in endometrial cancer. After the information provided in the corrected version, this objective has been clearly achieved.

Nevertheless, I still disagree with the affirmation that “the comparative analysis of EC patients with non-cancer patients with ascites, suggest (please correct as “suggests”) that theses (please correct as “these”) selected miRNAs come from EVs of EC tissue”. Authors cannot conclude this, since perhaps these differences simply reflect that the EVs-associated miRNA content is different in both pathologies, which can be influenced by several factors: the pathology itself, the source of the EVs (peritoneal lavage vs. ascitic fluids), surgery (after induction of general anaesthesia) and control samples obtained by means of paracentesis (with local anaesthesia).

From my point of view, a better and ethically acceptable control population could be peritoneal lavage in patients requiring surgery for leiomyomas or even peritoneal fluid from women requiring tubal ligation for definitive contraception. In this regard, the aforementioned confounding factors would be avoided. Authors may consider that the EVs-associated miRNAs between the proposed control population and the cirrhotic control population may vary in the same way that the results provided between EC and cirrhotic control population did, therefore preventing the conclusion that differences are only attributable to EC-associated miRNAs.

Nevertheless, and acknowledging the relevance of the study as the demonstration of the utility of peritoneal lavage as a source of EC-associeted miRNAs, I would recommend this paper for publication in the Cancers journal as far as authors include as a potential limitation of the study the composition of the control cohort. Finally, I would suggest adding a brief explanation in the discussion section on how the authors believe that specific EC-associated EVs  reach the peritoneal cavity in EC patients (perhaps through the Fallopian Tubes?)

Author Response

The manuscript “EV-associated miRNAs from peritoneal lavage is a source of biomarkers in endometrial cancer” (cancers-505924) evaluates the potential of peritoneal lavage during surgery in patients suffering from endometrial cancer as a source of biomarkers, in terms of EV-associated miRNAs. Authors found 8 miRNAs (namely miRNA-383-5p, miRNA-10b-5p, miRNA-34c-3p, miRNA-449b-5p, miRNA-34c-5p, miRNA-200b-3p, miRNA-2110 and miRNA-34b-3p), which demonstrated a classification performance at AUC values above 0.9 for patients with endometrial cancer. The paper is well written and English is appropriate.

After my initial comments, authors have provided more detailed information, explanations and performed the majority of suggested corrections regarding minor mistakes, which altogether have improved the quality of the paper.

On the one hand, a major limitation of the study in its initial form was the way in which samples had been processed, this is, freezing before centrifugations. Now that authors have clarified the right methodology, I have no further concerns in this point, which renders results more reliable. 

In their responses, the authors provided evidence that miRNA expression into their results was not affected by gender. Additionally, authors have provided further information for the patient cohort regarding histological grade and subtype, medical treatment, primary tumor localization and metastasis, which improved the quality of the paper. 

The authors highlight that the main objective of the paper is to demonstrate that peritoneal lavage is a source of EVs-related miRNAs in endometrial cancer. After the information provided in the corrected version, this objective has been clearly achieved. 

Point 1: Nevertheless, I still disagree with the affirmation that “the comparative analysis of EC patients with non-cancer patients with ascites, suggest (please correct as “suggests”) that theses (please correct as “these”) selected miRNAs come from EVs of EC tissue”. Authors cannot conclude this, since perhaps these differences simply reflect that the EVs-associated miRNA content is different in both pathologies, which can be influenced by several factors: the pathology itself, the source of the EVs (peritoneal lavage vs. ascitic fluids), surgery (after induction of general anaesthesia) and control samples obtained by means of paracentesis (with local anaesthesia). 

Response 1: We agree with the reviewer that we could not exclude the possibility that some of the miRNAs significantly differentiated in our study could be associated with other factors different from the EC pathology. As this is a limitation of our study, we have modified the text to highlight this limitation (line 248 to 256) as follows:

“The study has some open questions. This study permitted to identify a large number of dysregulated miRNAs associated with EVs in the peritoneal lavage of EC patients. We think that the EVs that we have identified come from endometrial and mesothelial cells in the EC group, whilst mostly from mesothelial cells in the control group. The comparative analysis of EC patients with non-cancer patients with ascites, suggests that these selected miRNAs come from EVs of EC tissue but there is obviously the probability that a subset of EVs came from an inflammatory reaction associated with EC and some other factors apart of the EC pathology, such as the source of EVs (peritoneal lavage vs. ascitic fluids), the surgery (after induction of general anaesthesia), and control samples obtained by means of paracentesis (with local anaesthesia)”.

Point 2: From my point of view, a better and ethically acceptable control population could be peritoneal lavage in patients requiring surgery for leiomyomas or even peritoneal fluid from women requiring tubal ligation for definitive contraception. In this regard, the aforementioned confounding factors would be avoided. Authors may consider that the EVs-associated miRNAs between the proposed control population and the cirrhotic control population may vary in the same way that the results provided between EC and cirrhotic control population did, therefore preventing the conclusion that differences are only attributable to EC-associated miRNAs. Nevertheless, and acknowledging the relevance of the study as the demonstration of the utility of peritoneal lavage as a source of EC-associeted miRNAs, I would recommend this paper for publication in the Cancers journal as far as authors include as a potential limitation of the study the composition of the control cohort.

Response 2: We agree with the reviewer that there exist other subgroups of patients that might be suitable to include in the control group. Nevertheless, our study was designed as a discovery phase study, in which we aim for the comparison of 2 groups of patients with a limited biologic variability in order to increase the chances of obtaining significant candidate biomarkers. We agree with the reviewer that in future studies, the biological variability of the control and case subgroups should be broader in order to eliminate false positive candidate biomarkers and validate our results. As the reviewer suggested, we agree that an appropriate control group of a validation phase could be the peritoneal lavage from women requiring surgery for leiomyomas or women requiring tubal ligation for definitive contraception. In order to explain this, we have included the next paragraph in the discussion section (line 256 to 264):

“Nevertheless, these promising biomarkers should be further validated as well as combined in order to increase the already excellent accuracy of each individual miRNAs. This should be done in an independent study involving a larger cohort of EC patients vs a control group with a higher biologic variability including, for exemple, patients with leiomyomas or women requiring tubal ligation for definitive contraception. Although we tested that differential miRNAs were not dependent on the gender factor (data not shown), further studies should include only female controls. Moreover, further research should be directed to evaluate the prognostic potential of each specific dysregulated miRNA as this might have implications to guide the surgical treatment of EC patients.”

Point 3: Finally, I would suggest adding a brief explanation in the discussion section on how the authors believe that specific EC-associated EVs reach the peritoneal cavity in EC patients (perhaps through the Fallopian Tubes?)

Response 3: We have add a sentence in the discussion (line 238 to 247) in order to clarify how the EC-associated EVs could spreach to the peritoneal cavity:

“There is great interest in the possibility that endometrial cancer cells may spread into the peritoneum and the ovaries through the Fallopian tube, and develop peritoneal metastasis and ovarian metastasis in the absence of lympho-vascular space invasion. It is important to take into account that normal endometrium may spread into peritoneum and the ovaries through retrograde menstruation, and give rise to endometriosis. Metastasis originated through this pathway may be associated with indolent behavior. A subset of ovarian metastasis from endometrial carcinomas are associated with such a good prognosis that they were interpreted a synchronous tumors [23]. Next generation sequence analysis confirms that they are metastatic. Assessing transtubal exosomal release from endometrial cancer may help in understanding the mechanisms involved in such type of indolent metastasis.”

23.       Schultheis, A.M.; Ng, C.K.Y.; De Filippo, M.R.; Piscuoglio, S.; Macedo, G.S.; Gatius, S.; Perez Mies, B.; Soslow, R.A.; Lim, R.S.; Viale, A.; et al. Massively Parallel Sequencing-Based Clonality Analysis of Synchronous Endometrioid Endometrial and Ovarian Carcinomas. JNCI.J 2015, 108, djv427.
